# I-Shaped Dimers of a Plant Chloroplast F_O_F_1_-ATP Synthase in Response to Changes in Ionic Strength

**DOI:** 10.3390/ijms241310720

**Published:** 2023-06-27

**Authors:** Stepan D. Osipov, Yury L. Ryzhykau, Egor V. Zinovev, Andronika V. Minaeva, Sergey D. Ivashchenko, Dmitry P. Verteletskiy, Vsevolod V. Sudarev, Daria D. Kuklina, Mikhail Yu. Nikolaev, Yury S. Semenov, Yuliya A. Zagryadskaya, Ivan S. Okhrimenko, Margarita S. Gette, Elizaveta A. Dronova, Aleksei Yu. Shishkin, Norbert A. Dencher, Alexander I. Kuklin, Valentin Ivanovich, Vladimir N. Uversky, Alexey V. Vlasov

**Affiliations:** 1Research Center for Mechanisms of Aging and Age-Related Diseases, Moscow Institute of Physics and Technology, 141700 Dolgoprudny, Russia; osipov.sd@phystech.edu (S.D.O.); rizhikov@phystech.edu (Y.L.R.); egor.zinoviev@gmail.com (E.V.Z.); minaeva.av@phystech.edu (A.V.M.); ivashchenko.sd@phystech.edu (S.D.I.); sudarev.vv@phystech.edu (V.V.S.); kuklina.dd@phystech.edu (D.D.K.); nikmicha@gmail.com (M.Y.N.); semenov.yury@gmail.com (Y.S.S.); 1989july@mail.ru (Y.A.Z.); i.s.okhrimenko@gmail.com (I.S.O.); gette.ms@phystech.edu (M.S.G.); dronova.ea@phystech.edu (E.A.D.); shishkin.aiu@phystech.edu (A.Y.S.); norbert_dencher@web.de (N.A.D.); alexander.iw.kuklin@gmail.com (A.I.K.); aleksei.vlasov@phystech.edu (A.V.V.); 2Frank Laboratory of Neutron Physics, Joint Institute for Nuclear Research, 141980 Dubna, Russia; 3Department of Molecular Medicine and Byrd Alzheimer’s Research Institute, Morsani College of Medicine, University of South Florida, Tampa, FL 33612, USA

**Keywords:** F_O_F_1_-ATP synthase, chloroplasts, dimers, small-angle scattering, membrane proteins

## Abstract

F-type ATP synthases play a key role in oxidative and photophosphorylation processes generating adenosine triphosphate (ATP) for most biochemical reactions in living organisms. In contrast to the mitochondrial F_O_F_1_-ATP synthases, those of chloroplasts are known to be mostly monomers with approx. 15% fraction of oligomers interacting presumably non-specifically in a thylakoid membrane. To shed light on the nature of this difference we studied interactions of the chloroplast ATP synthases using small-angle X-ray scattering (SAXS) method. Here, we report evidence of I-shaped dimerization of solubilized F_O_F_1_-ATP synthases from spinach chloroplasts at different ionic strengths. The structural data were obtained by SAXS and demonstrated dimerization in response to ionic strength. The best model describing SAXS data was two ATP-synthases connected through F_1_/F_1_′ parts, presumably via their δ-subunits, forming “I” shape dimers. Such I-shaped dimers might possibly connect the neighboring lamellae in thylakoid stacks assuming that the F_O_F_1_ monomers comprising such dimers are embedded in parallel opposing stacked thylakoid membrane areas. If this type of dimerization exists in nature, it might be one of the pathways of inhibition of chloroplast F_O_F_1_-ATP synthase for preventing ATP hydrolysis in the dark, when ionic strength in plant chloroplasts is rising. Together with a redox switch inserted into a γ-subunit of chloroplast F_O_F_1_ and lateral oligomerization, an I-shaped dimerization might comprise a subtle regulatory process of ATP synthesis and stabilize the structure of thylakoid stacks in chloroplasts.

## 1. Introduction

The ATP synthase has been the object of intensive studies for at least half a century. This interest is defined by the fact that the enzyme provides cells with adenosine triphosphate (ATP), which is the key molecule for almost all biochemical reactions in living organisms, being the major source of energy for use and storage at the cellular level. F-type ATP synthases were found in the mitochondria of eukaryotes, bacterial cellular membranes, and chloroplasts. ATP synthase couples transmembrane ion transport, established by mechanical rotation of a c-ring in a membrane (F_O_) region, with conformational changes in ATP catalytic centers at α/β interfaces (F_1_ region).

Mitochondrial ATP synthases (mtF_O_F_1_) are embedded in mitochondrial inner membranes and were found to form dimers of four different types (V-shape dimers of types I, II, and IV, and U-shape dimers of type III). These dimers differ in the angle between monomers of mtF_O_F_1_, which is responsible for the topology of mitochondrial cristae (special folds in the mitochondrial inner membrane) [1]. In contrast, in bacteria, ATP synthases (bF_O_F_1_) are known to be monomers [2]. ATP synthases in thylakoid membranes (cF_O_F_1_) were shown to be primarily monomeric with approx. 12% fraction of dimers and 3% of higher oligomers [3]. According to what is known, the contacts between two cF_O_F_1_ monomers are highly likely non-specific, and such dimerization might inhibit ATP synthases, preventing ATP hydrolysis in the absence of photosynthesis [4,5,6].

A dimerization of cF_O_F_1_ might be induced by changes in ionic strength in plant chloroplast during the light/dark cycles. The internal volume of a chloroplast is separated from the cytoplasm by a lipid membrane, which, together with the pumps and channels, makes it possible to create and maintain a difference between ion concentration inside and outside the chloroplast. This ion gradient can vary significantly from species to species, and between some organisms, it can reach more than ten-fold [7]. It is known that under light, various ions are redistributed in plants. The phenomenon of light-induced uptake of hydrogen ions by isolated chloroplasts is well studied. It is accompanied by the redistribution of other ions, such as Na, K, Cl, and Mg, implying that the ionic strength of the solution contained in plant chloroplasts can change 1.5 times during the day [8,9].

SAXS is an excellent structural method to study biological macromolecules in solutions, i.e., it allows one to perform the studies under a variety of conditions, including different ionic strengths. For example, recent studies show the possibility to obtain structural information about the supramolecular organization of membrane protein complexes from the experimental small-angle scattering (SAS) data, in particular, to determine the relative positions of monomers comprising an oligomer [10]. Accurate SAS data treatment for membrane proteins requires considering compounds surrounding the membrane part of the protein due to its significant influence on the I(Q) SAXS profile even for protein complexes with a large water-soluble part as was shown in the literature [11].

In our work, we observed the in vitro dimerization of cF_O_F_1_ in response to increasing ionic strength. The structural data were obtained by SAXS measurements of solubilized and purified samples of cF_O_F_1_ from spinach chloroplasts at different NaCl concentrations. SAXS data analysis resulted in a model of an I-shaped dimer of cF_O_F_1_, which is a mixture with monomeric cF_O_F_1_, providing the best fit of the experimental data. I-shaped co-linear F_1_/F_1_ interfaced cF_O_F_1_ dimers are formed by F_1_/F_1_ contacts, presumably via δ/δ subunit interaction, so that the F_O_ parts of the monomers are on the opposite sides of the dimer. We used a macromolecular docking to refine our model and improved the quality of the fit of SAXS data for each NaCl concentration thereby generating a high-resolution model of these dimers.

We present here a new model of a dimer of F-ATP synthases from spinach chloroplasts, which results in a specific F_1_/F_1_ interaction, presumably via δ-subunits, which connect a peripheral stalk (bb′) of ATP-synthase with the F_1_ catalytic head (α_3_β_3_), thus stabilizing it during rotation of the central stalk (γε) of the enzyme. We speculate that changes in ionic strength in plant chloroplasts during the day (light/dark cycles) might trigger the formation of cF_O_F_1_ oligomers in chloroplasts and, in particular, I-shaped cF_O_F_1_ dimers. We hypothesize that chloroplast ATP synthases might also form oligomers in order to inhibit ATP hydrolysis in the dark, and I-shaped dimers might possibly connect parallel opposing stacked thylakoid membrane areas and stabilize thylakoid stacks in chloroplasts.

The here reported model of the cF_O_F_1_ I-shaped dimer is unexpected and raises questions about whether such a type of dimerization exists in nature. First, such dimers should be inserted in two membranes at a specific distance from each other. Second, it should change the rate of ATP synthesis due to the steric hindrance of δ-subunits.

## 2. Results

### 2.1. Purification of the Monomeric F_O_F_1_ ATP-Synthase from Thylakoid Membranes

Isolation and purification of cF_O_F_1_ from spinach chloroplasts (Figure 1a) were conducted as described [12,13,14], and the final step of purification utilized treatment with the detergent 4-trans-(4-trans-Propylcyclohexyl)-cyclohexyl α-maltoside (t-PCCαM) followed by the anion exchange chromatography (AEX) against the NaCl concentration gradient (50–1000 mM) (Figure 1d). The AEX chromatogram showed a peak at ~300 mM NaCl (Figure 1e), which corresponds to the whole cF_O_F_1_ protein complex according to SDS-PAGE (Figure 1c). Blue-Native PAGE was also used to check the completeness and assembly of cF_O_F_1_ (Appendix A).

The peak fractions of cF_O_F_1_ were measured by SAXS and showed a good consistency (χ^2^ = 1.15) with the monomeric model of chloroplast ATP-synthase (PDB ID: 6FKF [15]) with a detergent belt built from the experimental SAXS data by MEMPROT program [16] (Figure 1b). Taking into account a detergent belt allows for improving the χ^2^ from 1.35 to 1.15 (Appendix A).

### 2.2. SAS Studies of cF_O_F_1_ Dimerization at Different NaCl Concentrations

We investigated the samples of purified cF_O_F_1_ by small-angle neutron scattering (SANS) in 100% H_2_O and 93% D_2_O buffers with 300 mM NaCl (for details see the section Small-angle scattering measurements in Materials and Methods). Obtained values of the maximum size (D_max_) and a radius of gyration (R_g_) were significantly more for the cF_O_F_1_ in the D_2_O buffer in comparison with the H_2_O one so that it could not be explained only by a contrast variation technique (Appendix A). This effect might have pointed towards an increase of an average oligomeric state in the sample, i.e., oligomerization polydispersity [11]. We hypothesized that a change in ionic strength could have led to the oligomerization of cF_O_F_1_. In order to verify this hypothesis, we measured samples of purified cF_O_F_1_ at a range of NaCl concentrations by small-angle X-ray scattering (SAXS) (Figure 2a).

It is worth mentioning that all samples of cF_O_F_1_ were first purified at the same conditions and after AEX the buffer contained 30 mM HEPES (pH 8.0), 2 mM MgCl_2_, 250–300 mM NaCl, 0.04% (*w*/*v*), tPCC-α-M (4-trans-(4-trans-Propylcyclohexyl)-cyclohexyl α-maltoside). Then, the samples were dialyzed against the same buffer with different NaCl concentrations (150, 250, 300, 350, 450 mM) and after dialysis, they were measured by SAXS.

Pair-distance distribution function P(r) for samples of cF_O_F_1_ at different NaCl concentrations (Figure 2b) showed the dependence of the maximum size of the object D_max_ (Figure 2c), radius of gyration Rg (Figure 2d), and Porod volume Vp (Figure 2e) on ionic strength, reaching the maxima at 150 and 450 mM NaCl and the minimum at 250–300 mM NaCl. It implies the increase of oligomerization fraction in the samples with both a decrease and an increase in ionic strength. Interestingly, the values D_max_ and R_g_ showed the lowest values in the samples after AEX, which then increase after dialysis against 250 and 300 mM NaCl (almost the same concentration of NaCl). It might indicate that the real NaCl concentration in the samples after AEX is slightly less than 300 mM NaCl due to estimation errors due to the mixing of the AEX peak fractions and other technical details of the chromatography. In addition, the real minimum of the values D_max_ and R_g_ might be between 250 and 300 mM NaCl.

The highest values of the fraction of cF_O_F_1_ dimers were observed for the lowest and the highest NaCl concentrations from investigated range (150 and 450 mM NaCl, respectively). The distance between two monomers of cF_O_F_1_ in the dimer can be estimated as follows:(1)$$D_{m-m}=2\sqrt{\frac{1+\alpha }{2\alpha }\left({Rg}_{mix}^{2}-{Rg}_{mono}^{2}\right)},$$
where *Rg_mono_* and *Rg_mix_* are radii of gyration for monomeric ATP-synthase and average for monomer-dimer mixure, correspondingly, *α* is a volume fraction of dimers estimated as *α* = *Vp_mix_*/*Vp_mono_* − 1 (for a detailed description see Appendix A). Equation (1) is based on the Guinier approximation, which is well-described in the literature [17]. The Porod volume Vp of the particle can be calculated using the Porod invariant Q [18]. Considering *Vp* errors as 10% and *Rg* errors (see Appendix A) we estimate *D_m-m_* equals to 183 ± 30 Å and 225 ± 24 Å for 150 mM and 450 mM NaCl, respectively.

The discrepancy of the values *D_m-m_* might indicate different dimerization mechanisms at low and high ionic strength. Nevertheless, such values of *D_m-m_* (similar to the size of cF_O_F_1_) imply that the contacts between two monomers of cF_O_F_1_ are at the most distant parts of these protein complexes. Thus, only two types of contacts are possible: F_O_/F_O_ (presumably via c-ring) or F_1_/F_1_ (presumably via δ-subunit).

### 2.3. Models of an I-Shaped cF_O_F_1_ Dimer

Taking into account the fact that the oligomerization increases at lower and higher ionic strength and the possible types of contacts derived from the *D_m-m_* value, we built two models of cF_O_F_1_ which satisfy these conditions. In the first case, we built a model comprising a mixture of a monomeric cF_O_F_1_ and dimers with F_O_/F_O_ contacts, presumably via c-ring (Figure 3a). In the second case, a model comprises a mixture of a monomeric cF_O_F_1_ and dimers with F_1_/F_1_ contacts, presumably via δ-subunit (Figure 3b).

The models of monomeric and dimeric cF_O_F_1_ contain detergent belts obtained from the experimental SAXS data and built by a program MEMPROT for the monomeric AEX fraction (see Figure 1b). The models of cF_O_F_1_ dimers were constructed using the HDOCK protein–protein docking webserver [19]. Two types of restraints have been used, providing F_O_/F_O_ or F_1_/F_1_ contacts, as described in the *Macromolecular docking* section in Materials and Methods.

An approximation of SAXS data with these models resulted in χ^2^ = 2.36 in the case of F_O_/F_O_ dimers and χ^2^ = 1.20 for F_1_/F_1_ dimers with a fraction of dimeric cF_O_F_1_ 96 ± 3% and 68 ± 2% for F_O_/F_O_ and F_1_/F_1_ dimers, respectively (Figure 3a,b). The approximation by the model containing F_1_/F_1_ dimers showed better χ^2^ value and relative residues ΔI/σ at small Q, which are within 3 (Figure 3d), in contrast to the approximation by the model containing F_O_/F_O_ dimers (Figure 3c). Therefore, the model comprising a mixture of a monomeric cF_O_F_1_ and dimers with F_1_/F_1_ contacts, presumably via δ-subunit, showed the best fit of the SAXS data.

In the case of the cF_O_F_1_ dimers with F_O_/F_O_ contacts, *D_m-m_* values for the top 10 HDOCK predictions (see *Macromolecular docking* section in Materials and Methods for details) were in the range of 174–201 Å. The only model (Figure 3a), which has *D_m-m_* ≥ 201 Å (a lower limit of the range 225 ± 24 Å) (Equation (1)), resulted in χ^2^ = 2.36 and a fraction of dimers ~96% for fitting of SAXS data for purified cF_O_F_1_ samples at 450 mM NaCl.

In the case of the cF_O_F_1_ dimers with F_1_/F_1_ contacts, *D_m-m_* values for the top 10 HDOCK predictions (see *Macromolecular docking* section in Materials and Methods for the details) were in the range of *D_m-m_* 150–210 Å. All of these models, including one shown in Figure 3b (Model 4 in Appendix A), in addition to contacts between δ-subunits, demonstrated contacts between other subunits, including α-, β-, b- and b′-subunit. Five out of ten models have *D_m-m_* ≥ 201 Å and demonstrated χ^2^ values in the range of 1.16–1.20 (see details in Appendix A) for fitting SAXS data for purified cF_O_F_1_ samples at 450 mM NaCl. Figure 3b shows one of these five models which corresponds to the best HDOCK confidence score (Appendix A). An additional validation of parameters of macromolecular interfaces estimated using the PDBePISA webserver [20] allowed us to asses this model as reasonable both in terms of HDOCK confidence score values and PDBePISA metric (see Appendix A for the details).

### 2.4. A Possible Physiological Role of I-Shaped cF_O_F_1_ Dimers

Concerning physiological relevance, we hypothesize that an I-shaped type of dimerization might stabilize the stacks of thylakoid membrane areas by the possible connection of neighboring lamellae. It is known that a large number of ATP-synthases is located in stroma lamellae and grana membranes [3,21] (Figure 4a). The high density of molecules in these membranes leads to the formation of contacts between them and lateral dimerization (Figure 4b). At the same time, the distance between neighboring stroma lamellae is about 30 nm, which approximately fits the I-shaped cF_O_F_1_ dimer (Figure 4c,d,g). It might also indicate a possibility of another type of dimerization via the F_1_ part of cF_O_F_1_, which is schematically shown in Figure 4e,f, and might have already been observed in the literature (Figure 4g).

## 3. Discussion

Studies of chloroplast ATP synthases already showed evidence of dimerization of cF_O_F_1_ from plant chloroplasts [3,4,5,6]. This dimerization is believed to be non-specific and is shown to be lateral, which means that two monomers of cF_O_F_1_ are interacting in one membrane. There is also evidence of cF_O_F_1_ dimers for single-cell green algae [4,5], however, the authors report only Blue Native PAGE and the following SDS PAGE, therefore no structural data have been reported and the exact type of this interaction remains unclear.

Using SAXS studies of purified samples of cF_O_F_1_ from spinach chloroplasts we showed that the degree of oligomerization increases in response to changes in the NaCl concentration, both a decrease and an increase from the concentration between 250 and 300 mM NaCl where the dimeric fraction is minimal. We expected to obtain a model of lateral interaction similar to that reported in the literature [3], but this model did not satisfy the distance between the two cF_O_F_1_ monomers (~200 Å) experimentally obtained by the SAXS technique. Thus, we came to a model with co-linear F_1_/F_1_ interfaced cF_O_F_1_ dimers, so that the F_O_ parts of the monomers are on the opposite sides of the dimer, having an I-shape if visualizing a detergent belt at the membrane F_O_ part of each monomer. Surprisingly, the model with F_O_/F_O_′ interaction did not fit well with the experimental SAXS data (χ^2^ = 2.3), although it satisfied the distance between the two cF_O_F_1_ monomers, in contrast to the model with F_1_/F_1_′ interaction.

Commonly, I(Q) SAXS profiles of membrane proteins solubilized in detergent, including large protein complexes, have a local maximum at 0.1–0.2 Å^−1^ [10,11]. This local maximum typically reflects the influence of a detergent belt on the I(Q) data and the presence of this maximum indicates the necessity of considering the detergent belt in the model of the protein complex for an accurate fit. However, our experiments showed another behavior of the I(Q) SAXS profile (see Figure 1d). Interestingly, SAXS data obtained for another type of ATP-synthase (A-type) (SASBDB ID: SASDKK4) [22] also showed the absence of a local maximum I(Q). Therefore, such behavior of the SAXS I(Q) profiles can be observed in the case of membrane protein complexes with a detergent belt, e.g., F- and A-type ATP-synthases.

We showed the first structural evidence of the I-shaped type of dimerization of chloroplast F-ATP synthases from plants. Our model suggests F_1_/F_1_′ contacts between the cF_O_F_1_ monomers as it is shown by SAXS data approximation (χ^2^ = 1.24) and it satisfies the value of the distance between the centers of masses of monomers $D_{m-m}$ (Equation (1)), which is about 200 Å. This means that the contacts between monomers of cF_O_F_1_ should be at the edge of the F_1_ part to satisfy the conditions of a comparatively large distance which is about the size of the whole protein complex cF_O_F_1_. Taking these facts together we hypothesize that the F_1_/F_1_′ contacts between cF_O_F_1_ monomers presumably might be via the δ-subunit, which is at the top of the catalytic head (α_3_β_3_) of the enzyme.

In order to check the possibility of the F_1_/F_1_ interaction we performed a macromolecular docking using the HDOCK protein–protein docking webserver [19]. Considering the δ-subunit as a key subunit for cF_O_F_1_ dimer formation (see *Macromolecular docking* section in Materials and Methods for the details) we obtained models of a dimer, which, in addition to contacts between δ-subunit, showed contacts between other subunits, including α-, β-, b- and b′-subunit. The resulting models can be assessed as reasonable both in terms of HDOCK confidence score values and parameters of macromolecular interfaces estimated using the PDBePISA webserver [20] (see Appendix A).

To understand which region(s) of a δ-subunit can potentially play a role in the I-shaped dimer formation, we evaluated the intrinsic disorder propensity of this protein, since disordered regions are known to often contribute to the protein-protein interactions, being capable of undergoing binding-induced folding [23,24,25,26]. Furthermore, even when a crystal structure of a protein of interest is solved, one often can find noticeable levels of intrinsic disorder, since a very significant fraction of proteins in PDB contains regions with missing electron density, which are potentially intrinsically disordered [27,28]. As we already indicated earlier [1], based on the analysis of the reported crystal structure of cF_O_F_1_ from the spinach chloroplasts (see PDB ID: 6FKF), each subunit of this protein contains regions of missing electron density, including the chain δ with residues 1–70 and 250–257. This indicates that very significant parts of each cF_O_F_1_ subunit are expected to be disordered even within the assembled complex [1]. The idea of the presence of significant levels of disorder in the cF_O_F_1_ δ-subunit is further supported by Figure 5a showing a model 3D-structure generated for the full-length δ-subunit by AlphaFold [29,30], which is the most accurate AI-based platform for the protein structure prediction [31] and Figure 5b representing the disorder profile of this protein generated based on the outputs of six commonly used disorder predictors from the PONDR family, such as PONDR^®^ VLXT [32], PONDR^®^ VL3 [33], PONDR^®^ VLS2 [34], and PONDR^®^ FIT [35], as well as IUPred2 (Short) and IUPred2 (Long) [36]. Figure 5 clearly shows that the long N-terminal region of the cF_O_F_1_ δ-subunit is expected to be highly disordered. Utilization of the flDPnn webserver that predicts disorder, disorder-based functions, and disordered linkers [37], revealed that residues 1–3, 22–26, and 59–66 of the cF_O_F_1_ δ-subunit might be involved in the disorder-dependent protein-protein interactions.

We should notice that it is a great challenge to show directly the I-shaped structural organization of cF_O_F_1_ because of weak non-specific interactions between the cF_O_F_1_ monomers. For example, a recent study demonstrated a cryo-EM structure of the monomer of ATP synthase from spinach chloroplasts reconstituted into lipid nanodiscs [15]. However, the procedure of grid preparation could expose the sample to harsh conditions, which might break the F_1_/F_1_′ contacts.

Another way of obtaining high-resolution structural information is protein crystallography. However, the crystallization of ATP synthase, as well as large membrane protein complexes, is the key challenge in structural biology nowadays [1]. A crystallization by using lipids *in meso* phases (*in meso* method), e.g., lipid cubic phases (LCP), provides the crystals of membrane proteins in close to native conditions [39]. However, due to limitations of the diameters of LCP water channels the size of a water-exposed part of the protein complex is limited. Potentially perspective can be *in bicelles* crystallization [40,41], which was successfully applied for several membrane proteins with a large water-soluble part [42,43,44]. However, at the moment of this writing, there are no reported successful cases of *in bicelles* crystallization of an ATP synthase.

ATP synthase crystallization by using the vapor diffusion (VD) method, which allows for avoiding the protein size limitations, has another problem connected with the stabilization of membrane proteins in solution and fast kinetics of crystallization processes. Thus, crystal structures of almost full ATP synthase complexes were obtained only in six cases during the whole period of ATP synthase studies [45,46,47,48,49,50], which indicates a great challenge in obtaining crystals of ATP synthase. It is worth pointing out that even if a crystal of cF_O_F_1_ is grown it can also rearrange contacts between cF_O_F_1_ monomers.

In order to observe I-shaped dimerization we used the SAXS method, which allowed us to obtain structural information about cF_O_F_1_ monomers and their supramolecular arrangement in close to native conditions. SAXS does not require crystals or low temperatures, although, in the case of a mixture of monomers and oligomers, SAXS data should be carefully analyzed [11,51].

We acknowledge that the presence of I-shaped dimers of chloroplast ATP-synthases is a hypothesis, which, ideally, should be directly checked, but one should be very careful about samples preparation and accurately control the ionic strength of the solution (e.g., for electron microscopy studies). Our structural data obtained by SAXS, though SAXS is an indirect method, point towards the possible presence of such dimers in purified samples of the cF_O_F_1_, which integrity was proven by BN-PAGE.

In literature, there are debates about the possible physiological roles of dimers and higher oligomers of cF_O_F_1_ [3,52]. Some papers show evidence of the presence of dimers of cF_O_F_1_ in microorganisms [4,5,53]. Other papers claimed that these dimers might be only aggregates without specific structural arrangement or functional role [3,52]. Commonly, inhibition of ATP hydrolysis in chloroplasts occurs via a redox switch inserted into a γ-subunit of cF_O_F_1_ [15], however, it is not excluded that there might be another way of ATP synthesis/hydrolysis regulation.

The formation of I-shaped dimers might establish an indirect interaction between lamellae (especially close to grana) and help to stabilize thylakoid stacks. Such dimers might be observed in more or less native conditions by cryo-electron tomography (Figure 4) and even the distance between neighboring lamellae, in principle, allows such interaction. However, reported structural studies of chloroplasts were made during the light phase [3,54], leaving a possibility to find native I-shaped dimers of cF_O_F_1_ in similar experiments at dark. When studying the mesoscale of thylakoids [3], I-shaped interactions could be indistinguishable from monomeric ATP synthases next to each other on adjacent lamellae. Of course, our hypothesis requires direct experimental verification, where the focus of the research will be directed specifically to the study of the protein supramolecular organization of lamellae and the search for mechanisms for their possible stabilization, as well as cryo-EM experiments with a focus on dimers or pairs of chloroplast ATP synthases.

Our study is intended to draw attention to a possible new type of ATP synthase dimerization, which has not been described previously, because regardless of the nature of such an interaction, we obtained experimental evidence of its presence (SAXS data), and given the fact that the ionic strength is comparable in order of magnitude in vitro and in vivo, especially given the fluctuations in ionic strength in plant chloroplasts during the light/dark cycles [8,9], we hypothesize that I-shaped dimerization may be reflected in native conditions and play a biological role, such as stabilizing thylakoid stacks or regulating ATP synthesis/hydrolysis in plants.

## 4. Materials and Methods

### 4.1. cF_O_F_1_ Isolation and Purification

The protein complex cF_O_F_1_ was isolated and purified from spinach chloroplasts as described [12,13] with minor modifications. Briefly, thylakoid membranes were solubilized in a buffer containing 60 mM n-Octyl-β-D-glucopyranoside and 25 mM Sodium Cholate *v*/*v* = 1/1, centrifuged at 200,000× *g* for 1 h and a pellet was discarded. Then the solution underwent ammonium sulfate (AS) precipitation in the range of 1.2–1.8 M of AS. The precipitate at 1.8 M AS was either resuspended in a buffer containing ~2 mM tPCC-α-M, centrifuged at 45,000× *g* and a pellet was discarded, a solution after 0.45 μm filtering was uploaded onto a column with POROS™ 20 HQ Strong Anion Exchange Resin and anion exchange chromatography was performed in a gradient of NaCl (50–1000 mM) in a buffer: 30 mM HEPES (pH 8.0), 2 mM MgCl_2_, 0.04% (*w*/*v*) tPCC-α-M; or alternatively precipitate at 1.8 M AS was resuspended in a buffer containing 16 mM N-dodecyl β-D-maltoside (DDM), centrifuged in a sucrose gradient (15–50% *w*/*v*) at 200,000× *g* for 16 h, and the fraction between 29 and 36% (*w*/*v*) of sucrose was desalted and underwent red-120 dye-ligand chromatography. Both methods provided ATP synthase applicable for further structural studies.

### 4.2. Blue Native Polyacrylamide Gel Electrophoresis

BN-PAGE was performed as described [55,56]. Briefly, the samples of cF_O_F_1_ were analyzed in linear gradient polyacrylamide gel (from 4 to 14% (*w*/*v*) BN-PAGE, the separation gel was overlayed with a 3% (*w*/*v*) sample gel). An electrophoresis apparatus (SE 400, Cytiva, Marlborough, MA, USA) was used with a cathode buffer containing 0.002% Coomassie brilliant blue G-250 (Bio-Rad, Hercules, CA, USA), phoresis duration was 16 h. All native electrophoresis runs were performed at 100 V and at 4 °C. Hight molecular weight markers (HMW, #17044501, Cytiva, Marlborough, MA, USA) were used with the 10 μL load on a line.

### 4.3. Small-Angle Scattering Measurements

SANS experiments were performed on the YuMO spectrometer (IBR-2, Dubna, Russia) [57] with a two-detector system [58,59]. For SANS measurements, two samples of cF_O_F_1_ were prepared. The first sample was obtained by anion-exchange chromatography (AEX). The second sample was obtained by dialysis of AEX-purified protein in 93% D_2_O-buffer with 300 mM NaCl, 30 mM HEPES (pH ~8.0 [60]), 2 mM MgCl_2_, 0.04% (*w*/*v*) tPCC-α-M. 93% is the final concentration of D_2_O in a sample solution of purified cF_O_F_1_ because of the comparable volumes of the sample and the dialysis buffer. Nevertheless, 93% of D_2_O was enough for obtaining a satisfactory incoherent background for SANS data treatment. Each sample (V = 400 μL) was poured into a Helma quartz cuvette (path length 1 mm) and placed in the temperature-controlled sample chamber [59] for further SANS measurements. The total exposure time for each sample was 80 min.

SAXS measurements were performed on the instrument Rigaku MicroMax-007 HF (MIPT, Dolgoprudny, Russia), which was used and described previously [41,61]. SAXS data were obtained for six samples of ATP synthase. The first sample with a protein concentration of ~5 mg/mL was obtained by purification using anion-exchange chromatography (see Figure 1d). Other five samples were obtained by overnight dialysis in buffers containing 30 mM HEPES, 2 mM MgCl_2_, 0.04% (*w*/*v*) tPCC-α-M, and different NaCl concentrations: 150, 250, 300, 350, and 450 mM. Each sample (V = 30 μL) was poured into a glass capillary, which was sealed by gas burner or wax and placed into a vacuum chamber at a distance of 2.0 m from a multiwire gas-filled detector Rigaku ASM DTR Triton 200. All measurements were performed at room temperature. The thermal stability of cF_O_F_1_ at room temperature was checked by Dynamic Scanning Fluorimetry (see Section 4.6). See Appendix A for other details of SAXS measurements and data treatment.

### 4.4. Small-Angle Scattering Data Treatment

In the case of SAXS data, first, SAXSGui v. 2.15.01 software was used to convert 2D images of scattering intensity vs. transmitted momentum in reciprocal space Q = 4π sin(θ)/λ (where λ = 1.5405 Å (K_α_ of Cu), 2θ-a scattering angle) to 1D-profiles I(Q) by using a radial integration. For Q-calibration, a powder of silver behenate (AgBh) was used (Q = 0.1076 Å^−1^, d = 58.38 Å) [62]. In the case of SANS data, raw data were converted to one-dimensional data set I(Q) with a program SAS (version 5.1.5) [63]. Then, in both cases, a regularized model fit of I(Q) 1D-profiles was used for the calculation of a pair-distance distribution function P(R) by a program GNOM from the ATSAS v. 2.8.5 program package [64,65]. Visualizing of high-resolution models of cF_O_F_1_ dimers was performed by PyMOL v. 1.9 software [66]. MEMPROT v. 2.2 software [16] was used to fit an experimental SAXS data for AEX-purified cF_O_F_1_ using a high-resolution model of cF_O_F_1_ from spinach chloroplasts (PDB ID: 6FKF) with a pseudo-atomic model of the detergent belt surrounding its transmembrane part. For proper orienting of the cF_O_F_1_ model before running MEMPROT v. 2.2 (place the center of the transmembrane part at the origin (zero) and set the normal vector to the membrane plane along the *z*-axis) we used a PPM web server v. 2.0 [67]. A program OLIGOMER [68] from ATSAS v. 2.8.5 was used to fit experimental SAXS data from a two-component mixture of monomers and dimers of cF_O_F_1_ and to obtain the volume fractions of each component in the mixture. A program CRYSOL v.2.0 [69] (command line mode) from ATSAS v. 2.8.5 was used for evaluating the solution scattering from macromolecules in order to fit experimental SAXS data and/or prepare a set of form-factors for the subsequent run of the OLIGOMER program from ATSAS v. 2.8.5. See Appendix A for other details of SAXS measurements and data treatment.

### 4.5. Macromolecular Docking

For validation of possible F_1_/F_1_ interaction of cF_O_F_1_ monomers, we performed a macromolecular docking using HDOCK protein–protein docking webserver [70]. Considering the δ-subunit as a key subunit for dimer formation observed by SAXS, we used the following residue distance restraints: 1–257:d 1–257:d 50. Here, the letter “d” corresponds to the δ-subunit chain, “1–257” are numbers that cover all residues presented in δ-subunit, and the number “50” corresponds to the condition that residues 1–257 of a chain “d” on the receptor and on the ligand will be within 50 Å. Screening of the distance value in the range of 10–150 with a step = 10 shows the same docking results. The top 10 obtained HDOCK models, in addition to contacts between δ-subunit, demonstrated contacts between other subunits, including α, β, b, and b′, and showed the values of the HDOCK confidence score in the range 0.39–0.52 (Appendix A). Accordingly, to the description of HDOCK, it is considered that two molecules would be possible to bind when the confidence score is between 0.5 and 0.7. However, taking into account the comment of the authors: “the confidence score here should be used carefully due to its empirical nature”, the lower value of 0.5 is empirical and models with scores slightly below 0.5 can also be considered as possible and reasonable. Additionally, we checked the parameters of macromolecular dimerization interfaces using the PDBePISA webserver [20]. Results are shown in Appendix A.

In order to check the possibility of F_O_/F_O_ interaction we also performed a macromolecular docking using an HDOCK webserver. Considering the c-subunit as a key subunit for dimer formation observed by SAXS, we used the residue distance restraints corresponding to 10 Å distance between residue 3 in c-subunits on the receptor and on the ligand (3:g 3:q 10). The only model that satisfies the condition *D_m−m_* ≥ 201 Å (Equation (1)), has a high value of the HDOCK confidence score = 0.78 (Docking Score = −212.65), which corresponds to a high probability of the formation of such a contact. However, dimerization via F_O_/F_O_ interaction was not confirmed using SAXS data.

### 4.6. Dynamic Scanning Fluorimetry

We used Prometheus Panta from NanoTemper Technologies (München, Germany) to check the thermostability of the protein complex cF_O_F_1_. Experiments were carried out in standard capillaries Prometheus NT.48 (Cat. # PR-C002). A temperature scan was conducted with a 4 °C/min slope for a temperature range of 15–75 °C, an excitation LED power was 70%. Thermal stability parameters (T_onset_, T_m_) were calculated by Panta Analysis software v. 1.2 (NanoTemper Technologies, München, Germany). Folding state transition was monitored by the ratio of fluorescence intensity at 330 nm and 350 nm as a function of temperature, where T_onset_ (onset temperature of thermal unfolding) was 51.6 °C and T_m_ (inflection temperature of thermal unfolding) was 66.9 °C (Appendix A). Thus, at 20 °C the stability of the protein was not disrupted.

## Figures and Tables

**Figure 1 ijms-24-10720-f001:**
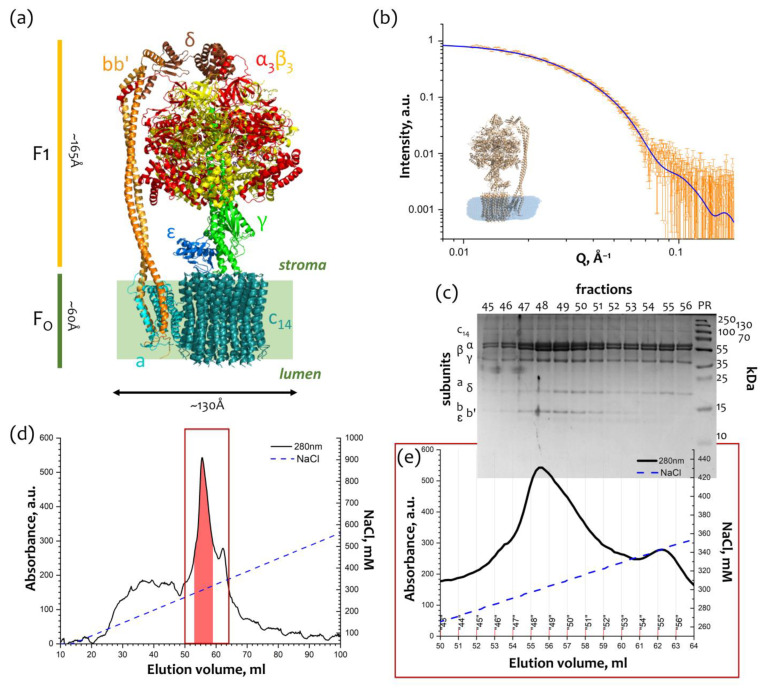
Purification and characterization of cF_O_F_1_: (**a**) Overall view of ATP synthase from spinach chloroplasts. F_O_ part consists of abb′c_14_ subunits, F_1_ part consists of α_3_β_3_γεδ subunits and bb′ subunits form a peripheral stalk. The structure of ATP synthase from spinach chloroplasts (PDB ID: 6FKF [15]) was used for representation; (**b**) Characterization of the peak fractions of cF_O_F_1_ after anion exchange chromatography (AEX) by SAXS, experimental I(Q) 1D-profile showed as hollow grey stars, a model of cF_O_F_1_ (PDB ID: 6FKF) with a detergent belt was used for approximation of experimental data with χ^2^ = 1.15 (blue line); (**c**) Characterization of the AEX peak fractions of cF_O_F_1_ by SDS PAGE, colored with Coomassie; (**d**) Anion exchange chromatography (AEX) of cF_O_F_1_ from spinach chloroplasts, highlighted fractions were merged and taken for structural studies, the region shown with a red box is described in details in Panel (**e**); (**e**) AEX peak fractions of cF_O_F_1_. Panels (**a**,**c**–**e**) were adapted from [12] with modifications.

**Figure 2 ijms-24-10720-f002:**
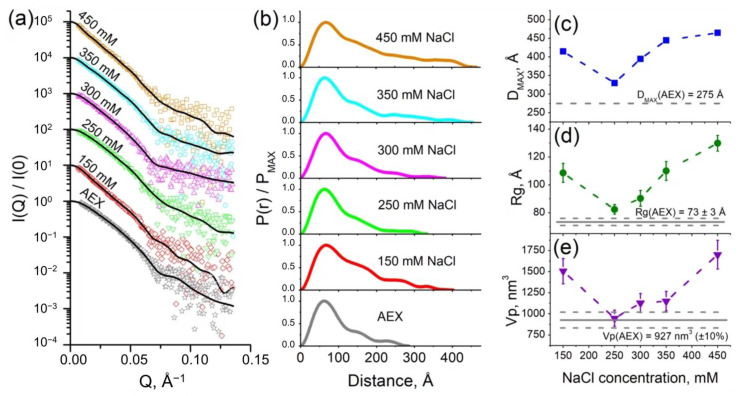
SAXS characterization of cF_O_F_1_: (**a**) Experimental I(Q) 1D-profiles for AEX-purified and dialyzed against the same buffer with different NaCl concentrations samples of cF_O_F_1_, experimental data are shown as dots, regularized fits are shown as black lines. For clarity, SAXS data for 150, 250, 300, 350, and 450 mM NaCl were multiplied by 10, 10^2^, 10^3^, 10^4^, and 10^5^, respectively; (**b**) Normalized pair-distance distribution function P(r) for cF_O_F_1_ at different NaCl concentrations; (**c**) Maximum size of the object (D_max_) for samples of cF_O_F_1_ at different NaCl concentrations; (**d**) Radius of gyration (Rg) for samples of cF_O_F_1_ at different NaCl concentrations obtained from P(r); (**e**) Porod volume (Vp) for samples of cF_O_F_1_ at different NaCl concentrations. Gray lines show the values of D_max_, Rg, Vp for an AEX-purified sample of cF_O_F_1_ without dialysis. The values used for plots in panels (**c**–**e**) are given in Appendix A.

**Figure 3 ijms-24-10720-f003:**
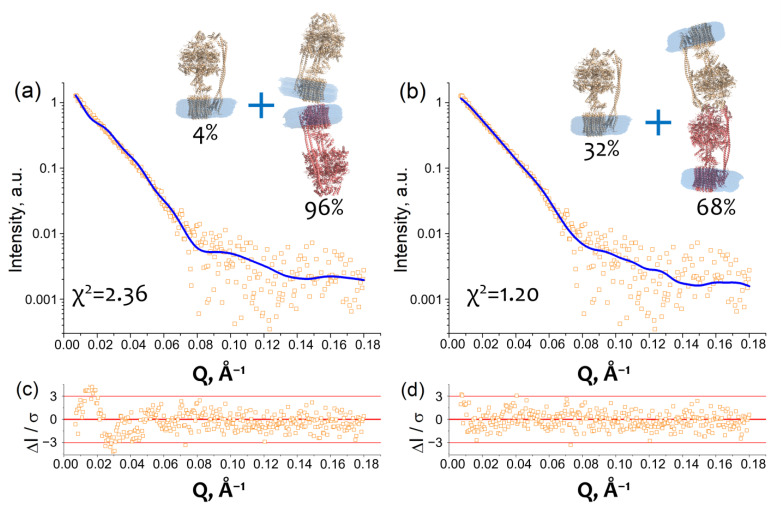
Dimerization of cF_O_F_1_ from Spinacia oleracea at 450 mM NaCl shown by SAXS: (**a**) Experimental I(Q) 1D-profile for cF_O_F_1_ at 450 mM NaCl (orange squares) and an approximation (blue line, χ^2^ = 2.36) by using a model of a mixture of cF_O_F_1_ monomers and dimers formed by F_O_-F_O_ contacts (see *Macromolecular docking* section in Materials and Methods); (**b**) Experimental I(Q) 1D-profile for cF_O_F_1_ at 450 mM NaCl (the same as in panel (**a**), orange squares) and an approximation (blue line, χ^2^ = 1.20) by using a model of a mixture of cF_O_F_1_ monomers and dimers formed by F_1_-F_1_ contacts (see *Macromolecular docking* section in Materials and Methods); (**c**) Relative residues of the fit shown in panel (**a**); (**d**) Relative residues of the fit shown in panel (**b**). The models of monomeric and dimeric cF_O_F_1_, shown in panels (**a**,**b**), contain detergent belts obtained from the experimental SAXS data and built by the program MEMPROT. Volume fractions of monomers and dimers of cF_O_F_1_, used in the models for approximation, are shown.

**Figure 4 ijms-24-10720-f004:**
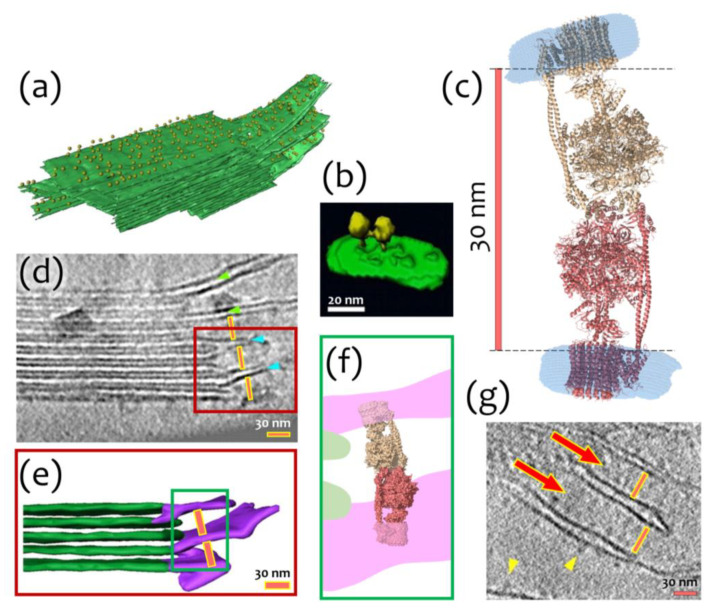
Possible biological interpretation of observed cF_O_F_1_ I-shaped dimer: (**a**) Segmented subvolume of a thylakoid stack with connected stroma lamellae, cF_O_F_1_ are shown as yellow 12-nm spheres, cF_O_F_1_ is randomly distributed over stromal lamellae; (**b**) Surface rendering of a lateral cF_O_F_1_ dimer in isolated pea thylakoid membranes; (**c**) A model of an I-shaped dimer with F_1_/F_1_ interaction interface, presumably via δ-subunit, the distance between membrane F_O_ parts of different cF_O_F_1_ monomers is about 30 nm; (**d**) Electron Tomography of Vitreous Spinach Chloroplast Sections, stacked grana and unstacked stroma thylakoid membranes are shown, the distance between neighboring stroma lamellae approximately fits the size of the I-shaped cF_O_F_1_ dimer, the stroma thylakoids, which are continuous with a grana thylakoid is shown with green arrowheads, the stroma thylakoids merged with two adjacent grana thylakoids—blue arrowheads, the region shown with a red box is described in details in Panel (**e**); (**e**) Surface representation of connections between grana (green) and stroma (purple) thylakoids, the distance between neighboring stroma lamellae is about 30 nm, the region shown with a green box is described in details in Panel (**f**); (**f**) Schematic representation of an I-shaped cF_O_F_1_ dimer in between of two neighboring stroma lamellae; (**g**) A part of a tomographic slice showing neighboring stroma lamellae, the distance between them approximately fits the size of an I-shaped ATP-synthase dimer, Pink lines are 30 nm size and demonstrate the possible fit of an I-shaped cF_O_F_1_ dimer between the neighbor lamellae. Presumably, ATP synthases shown with red arrows might be these I-shaped dimers. Yellow arrowheads show separate cF_O_F_1_. Panels (**a**,**b**,**d**,**e**,**g**) were adapted from [3] with modifications.

**Figure 5 ijms-24-10720-f005:**
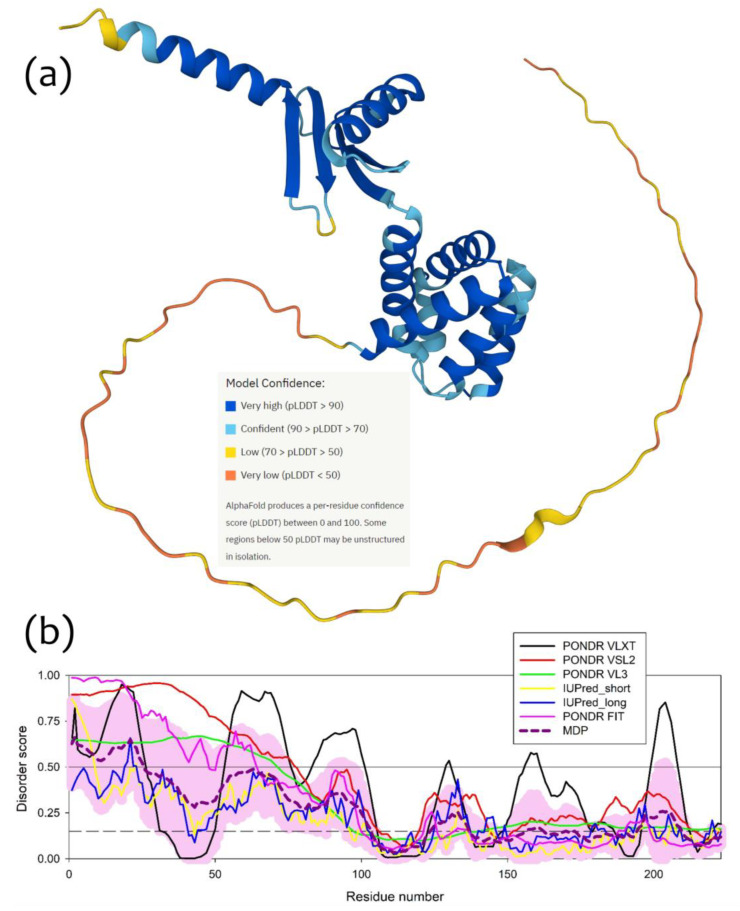
Structure and disorder in the cF_O_F_1_ δ-subunit: (**a**) 3D structure of the cF_O_F_1_ δ-subunit generated by AlphaFold2; (**b**) Per-residue intrinsic disorder predisposition of the cF_O_F_1_ δ-subunit from the spinach chloroplasts. The intrinsic disorder profile was generated using the outputs of the Rapid Intrinsic Disorder Analysis Online (RIDAO) platform [38] that aggregates the results from a number of well-known disorder predictors, such as PONDR^®^ VLXT, PONDR^®^ VL3, PONDR^®^ VLS2, PONDR^®^ FIT, and IUPred2 (Short) and IUPred2 (Long) and also produces the mean disorder profile (MDP) and corresponding error distribution (pink shadow). The outputs of the evaluation of the per-residue disorder propensity by these tools are represented as real numbers between 1 (ideal prediction of disorder) and 0 (ideal prediction of order). A threshold of 0.5 is used to identify disordered residues and regions in query proteins. Residues with the disorder scores (DS) DS < 0.15 are considered as ordered, residues with 0.15 ≤ DS < 0.25 are taken as flexible, whereas residues with 0.25 ≤ DS < 0.5 are considered as moderately disordered.

## Data Availability

SAXS data of cF_O_F_1_ at different NaCl concentrations are deposited in SASBDB with the IDs: SASDRR8, SASDRS8, SASDRT8, SASDRU8, SASDRV8, SASDRW8.

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
