# Peer review of "I-Shaped Dimers of a Plant Chloroplast FOF1-ATP Synthase in Response to Changes in Ionic Strength"

_ijms, 2023, doi:10.3390/ijms241310720_

Round 1

Reviewer 1 Report

In this work CF0F1 was isolated from spinach chloroplasts and the resulting enzyme preparation was investigated by small-angle X-ray scattering at different ionic strengths (150-450 mM NaCl). The SAXS characterization of these samples give the pair distance distribution, the maximum size of the object and the radius of gyration  as function of the NaCl concentration. These three parameters show maxima at 150 mM and 400 mM and a minimum at 250 mM NaCl. The interpretion is as follows:  with increasing  ionic strength first the oligomerisation of CF0F1  decreases, reaches a minimum and then increases again. The authors explain this with a dimerization of CF0F1 . In the following they discuss two model dimers with F0/ F0 contacts (via c-subunits) and dimers with F1/ F1 contacts (via delta subunits).

I have two problems with this work:

1. The samples for the SAX measurement are carried out with CF0F1 in the AEX buffer (line136-138 in the manuscript) and then dialyzed against  different NaCl concentrations. Unfortunately, the dialysis membrane and the process is not described in detail in this work. If a usual dialysis membrane is used, during this process all small ions (HEPES, MgCl2, detergent) are lost und substituted by NaCl. This leads to different effects: the loss of detergent results in an oligomerisation of  the enzyme, the loss of buffer and MgCl2 result in denaturation of the enzyme and finally, the increasing NaCl concentration might influence the state of  the enzyme. It is difficult to interprete results obtained with  samples when the state of enzyme is changed by different processes.

2. It is known from electronmicroscpic data that monomers, dimers and oligomers can be observed in samples of isolated CF0F1 and CF0 (see ref. 49). The aggregation occurs in most cases via side side contact of the CF0 parts and in ref. 49 the diameter of CF0  and the thickness of the detergent belt of CF0F1 are reported. The formation of dimers as proposed in Fig.3 a and b in this work was not observed. It is not clear why the authors prefer to model the detergent belt with a program and do not use experimental data.

Fig. 4 shows data from ref. 3 and in addition the dimer model from Fig. 3 b is shown.  It is claimed that the distance between neighboring stroma lamella is determined by the size of this CF0F1 dimer localized in two different stroma lamella. However, in ref.3 it is stated that CF0F1 occurs in the membranes as monomer and any association with two or more  complexes are due to stochastic contacts.

Author Response

Comments and Suggestions for Authors

In this work CF0F1 was isolated from spinach chloroplasts and the resulting enzyme preparation was investigated by small-angle X-ray scattering at different ionic strengths (150-450 mM NaCl). The SAXS characterization of these samples give the pair distance distribution, the maximum size of the object and the radius of gyration  as function of the NaCl concentration. These three parameters show maxima at 150 mM and 400 mM and a minimum at 250 mM NaCl. The interpretion is as follows:  with increasing  ionic strength first the oligomerisation of CF0F1  decreases, reaches a minimum and then increases again. The authors explain this with a dimerization of CF0F. In the following they discuss two model dimers with F0/ F0 contacts (via c-subunits) and dimers with F1/ Fcontacts (via delta subunits).

I have two problems with this work:

  1. The samples for the SAX measurement are carried out with CF0F1in the AEX buffer (line136-138 in the manuscript) and then dialyzed against  different NaCl concentrations. Unfortunately, the dialysis membrane and the process is not described in detail in this work. If a usual dialysis membrane is used, during this process all small ions (HEPES, MgCl2, detergent) are lost und substituted by NaCl. This leads to different effects: the loss of detergent results in an oligomerisation of  the enzyme, the loss of buffer and MgCl2result in denaturation of the enzyme and finally, the increasing NaCl concentration might influence the state of  the enzyme. It is difficult to interprete results obtained with  samples when the state of enzyme is changed by different processes.

Answer: Thank you for your remark. A dialysis was performed against a buffer that also contained HEPES, MgCl2 and a detergent in the same amount as the initial protein solution, the difference was only in a concentration of NaCl. We slightly modified the text and the legend to the fig. 2 for clarity.

  1. It is known from electronmicroscpic data that monomers, dimers and oligomers can be observed in samples of isolated CF0F1and CF0(see ref. 49). The aggregation occurs in most cases via side side contact of the CF0 parts and in ref. 49 the diameter of CF and the thickness of the detergent belt of CF0Fare reported. The formation of dimers as proposed in Fig.3 a and b in this work was not observed. It is not clear why the authors prefer to model the detergent belt with a program and do not use experimental data.

Answer: We thank the reviewer for this valuable comment.

First, unfortunately the data reported in the ref. 49 (ref. 52 in new version) are not sufficient to build a model of a detergent belt that is required for calculation of SAXS I(Q) profile. It is necessary to mention that one should carefully build a model of a detergent belt due to its significant influence on the I(Q) SAXS profile [see ref. 11]. For accurate calculation of the SAXS profile from a detergent belt one should know not only the outer sizes but also some details about the internal structure, namely all parameters: a, b, t, e and φ (see description in MEMPROT-related ref. 16, in particular Fig. 1 in ref. 16). Moreover, the parameters of a detergent belt depend on the detergent compound. In our case a detergent belt of the purified ATP-synthase consists of 4-trans-(4-trans-Propylcyclohexyl)-cyclohexyl α-maltoside (t-PCCαM), which has different properties in comparison to the membrane-mimicking compounds used, for instance, for sample preparation of ATP-synthases shown in ref. 49 (ref. 52 in new version).

Second, the conditions during sample preparation for electron microscopy (EM) studies can significantly differ from native conditions. In particular, the ionic strength of a sample can be changed uncontrollably during sample preparation for EM, which might lead to protein aggregation. SAXS data were obtained from purified proteins which integrity was confirmed by BN-PAGE, therefore the only varied parameter was NaCl concentration. Consequently, the aggregation was connected with controllable changing of ionic strength and our experimental SAXS data showed the form-factor close to the models described in the manuscript (I-shaped dimers).

Finally, we used the experimental SAXS data to build a model of a detergent belt. We used the program MEMPROT which has proven its efficiency in building models of membrane proteins with detergent belts [ref. 16]: it allows one to select the parameters of a detergent belt towards the best fit of the experimental SAXS data by a model I(Q) SAXS profile to (i.e., minimize χ2). In our work, the parameters of the t-PCCαM detergent belt, that stabilizes the cFOF1, obtained from the experimental SAXS data are described in the Table S1e.

Accordingly, we modified the text and the legend of the fig. 3 for clarity.

Fig. 4 shows data from ref. 3 and in addition the dimer model from Fig. 3 b is shown.  It is claimed that the distance between neighboring stroma lamella is determined by the size of this CF0Fdimer localized in two different stroma lamella. However, in ref.3 it is stated that CF0F1 occurs in the membranes as monomer and any association with two or more  complexes are due to stochastic contacts.

Answer: We thank the reviewer for this notice.

The F1-F1 contacts are really reported as stochastic in ref. 3. However, it doesn’t contradict to the possible presence of I-shaped dimers which can be formed at different NaCl concentrations. We acknowledge that I-shaped dimers of chloroplast ATP-synthases is a hypothesis, which, ideally, should be directly checked, but one should be very careful about samples preparation and accurately control the ionic strength of the solution (e.g., for electron microscopy studies). Our structural data obtained by SAXS, though SAXS is an indirect method, point towards the possible presence of such dimers in purified samples of the cFOF1, which integrity was proven by BN-PAGE. Finally, the fact that we observed dimerization of chloroplast ATP synthases in response to changes in ionic strength is of interest regardless of the nature of this interaction, since it may reflect native conditions, especially given the fluctuations in ionic strength in plant chloroplasts during the light/dark cycles. This may represent a new way of fine-tuning of ATP synthesis/hydrolysis in plants or stabilization the structure of thylakoid stacks.

The expanded description is added in the end of the Discussion section .

Reviewer 2 Report

The article was a very interesting read, I think it would be worthy to be published in the International Journal of Molecular Sciences after several minor and some major improvements. Please see below the list of corrections asked from the authors.

Line 46: Please introduce shortly the mitochondria cristae

Line 77: Please introduce shortly the delta subunits

Introduction: Please discuss shortly prior relevant SAS works on relevant systems

Figure 1: the order of subfigures is a, c, d, b. Please rearrange in a, b, c, d order.

Lines 108 – 112 are phrased in a very unclear fashion, please improve the clarity of the message.

 Line 116. Why was 93 % D2O chosen and not 100 %. Please comment.

Lines 115-117 is unclear. Increase as compared to what? Why was it expected? Please clarify.

Figure 2. Please comment why AEX is so different from the 300 mM data.

Lines 140-145. Confusing sentence, please clarify. Increase of a parameter with changing of ionic strength would make the reader assume a monotonic function, which is not the case.

Line 146. When the reader reads this sentence, it is unclear why you assume dimerization most certainly.

Equation 1) Either provide a reference for this or provide a detailed explanation in the SI.

Lines 150-151. Define Alpha, the different Vp values, and why this unconventional ratio was used to estimate volume fraction of dimers.

Lines 267-268. Please show a 3D rendering of the complex and highlight the different subunits.

Lines 303-305 are wrong. Please revise.

Line 326 Please revise “only in several”

Lines 342-349. The conclusion is a bit rudimentary. There are existing structural studies related to illumination, ionic strength etc. with a range of different techniques (including EM, SAXS and SANS) with direct information about the stacking, order, and in general the mesoscale structure of the thylakoids. They may or may not agree with your proposed hypothesis. Please perform a detailed critical discussion about how your hypothesis may fit into the framework of prior published results.

Figure S2 legend. Please use subscript in cFOF1

Figure S2 and S3 are cited in reverse order, please rearrange.

The quality of English Language is relatively good. Some sentences should be clarified as detailed in the list above.

Author Response

Comments and Suggestions for Authors

The article was a very interesting read, I think it would be worthy to be published in the International Journal of Molecular Sciences after several minor and some major improvements. Please see below the list of corrections asked from the authors.

Line 46: Please introduce shortly the mitochondria cristae

Answer: We thank the reviewer for the remark. We added the short sentence about mitochondria cristae and their connection with mtFOF1:

“These dimers differ in the angle between monomers of mtFOF1, which is responsible to the topology of mitochondrial cristae (special folds in mitochondrial inner membrane) [1].”

Line 77: Please introduce shortly the delta subunits

Answer: We added a short description of the delta subunits in the text:

“δ-subunits, which connect a peripheral stalk (bb’) of ATP-synthase with the F1 catalytic head (α3β3), thus stabilizing it during rotation of the central stalk (γε) of the enzyme (Figure 1a).”

Introduction: Please discuss shortly prior relevant SAS works on relevant systems

Answer: Thank you for this comment, we agree that this should be given. We added a short description of prior relevant SAS works on large membrane protein complexes to the Introduction:

“For example, recent studies show the possibility to obtain the structural information about supramolecular organization of membrane protein complexes from the experimental small-angle scattering (SAS) data, in particular, to determine the relative positions of monomers comprising an oligomer [7]. Accurate SAS data treatment for membrane proteins requires considering compounds surrounding the membrane part of the protein due to its significant influence on the I(Q) SAXS profile even for protein complexes with a large water-soluble part as it was shown in literature [8].”

Figure 1: the order of subfigures is a, c, d, b. Please rearrange in a, b, c, d order.

Answer: We rearranged the subfigures and added a subfigure showing a 3D rendering of the ATP-synthase with highlighted different subunits.

Lines 108 – 112 are phrased in a very unclear fashion, please improve the clarity of the message.

Answer: We agree that this sentence should be written clearer. We modified it, added more description about the connection between I(Q) SAXS profile and a detergent belt and moved this text into Discussion:

“Commonly, I(Q) SAXS profiles of membrane proteins solubilized in detergent, including large protein complexes, have a local maximum at 0.1 – 0.2 Å-1 [7,8]. This local maximum typically reflects the influence of a detergent belt on the I(Q) data and the presence of this maximum indicates the necessity of considering the detergent belt in the model of the protein complex for an accurate fit. However, our experiments showed another behavior of the I(Q) SAXS profile (see the Fig.1d). Interestingly, SAXS data obtained for another type of ATP-synthase (A-type) (SASBDB ID: SASDKK4) [19] also show the absence of a local maximum I(Q). Therefore, such behavior of the SAXS I(Q) profiles can be observed in case of membrane protein complexes with a detergent belt, e.g., F- and A-type ATP-synthases.”

 Line 116. Why was 93 % D2O chosen and not 100 %. Please comment.

Answer: We thank the reviewer for this comment and we agree that additional explanation is required here.

93% is the final concentration of D2O in a sample solution of purified cFOF1 because of the comparable volumes of the sample and a dialysis buffer: D2O, 300 mM NaCl, 30 mM HEPES (pH ~8.0 [59]), 2 mM MgCl2, 0.04 % (w/v) tPCC-α-M. Nevertheless, the 93% of D2O was enough for obtaining a satisfactory incoherent background for SANS data treatment.

Lines 115-117 is unclear. Increase as compared to what? Why was it expected? Please clarify.

Answer: Thank you for improving the clarity of our manuscript. We added the explanation in the text:

“We investigated the samples of purified cFOF1 by small-angle neutron scattering (SANS) in 100% H2O and 93% D2O buffers with 300 mM NaCl (for details see the section Small-angle scattering measurements in Materials and Methods). Obtained values of the maximum size (Dmax) and a radius of gyration (Rg) were significantly more for the cFOF1 in the D2O buffer in comparison with the H2O one, so that it could not be explained only by a contrast variation technique (Figure S2).”

Figure 2. Please comment why AEX is so different from the 300 mM data.

Answer: We thank the reviewer for this important notice.

The point is that when doing the anion exchange chromatography in a gradient of NaCl we can only approximately estimate the final concentration of NaCl in the sample. According to the chromatogram (fig. 1d, e) the concentration should be approx. 300 mM, however, the error of this estimation is about 10% due to mixing of peak fractions and other technical details of the chromatography. Taking into account that Dmax and Rg have minimal values at approx. 250mM NaCl, we assume that 1) the real AEX concentration might be slightly less than 300 mM, and 2) Dmax and Rg values might reach a real minimum in between of 250 – 300 mM NaCl. It might explain the lowest values of Dmax and Rg in AEX samples in comparison to those for samples at 300 mM and even at 250 mM NaCl concentrations after dialysis.

We also added the explanation in the text in the section 2.2. “SAS studies of cFOF1 dimerization at different NaCl concentrations”.

Lines 140-145. Confusing sentence, please clarify. Increase of a parameter with changing of ionic strength would make the reader assume a monotonic function, which is not the case.

Answer: Thank you for the remark. We made corrections in the text and changed the “Increase of a parameter with changing of ionic strength” to “Dependence of a parameter on changing of ionic strength”.

Line 146. When the reader reads this sentence, it is unclear why you assume dimerization most certainly.

Answer: We thank the reviewer for this comment. We meant “The highest values of the fraction of cFOF1 dimers were observed...” We changed the text so that make it clear.

Equation 1) Either provide a reference for this or provide a detailed explanation in the SI.

Answer: We added the detailed description of the Equation 1 in the Text document S1 in Supplementary materials and provided a link in the text.

Lines 150-151. Define Alpha, the different Vp values, and why this unconventional ratio was used to estimate volume fraction of dimers.

Answer: We appreciate the reviewer for noticing this misprint. Of course, Alpha = Vpmix/Vpmono – 1. We fixed the misprint in the text and also described in detail the derivation of this formula in Text document S1 in Supplementary materials.

Lines 267-268. Please show a 3D rendering of the complex and highlight the different subunits.

Answer: We added a subfigure (fig. 1a) showing a 3D rendering of the ATP-synthase with highlighted different subunits.

Lines 303-305 are wrong. Please revise.

Answer: Thank you for this notice. We revised the text and now it is the following: “Residues with the disorder scores (DS) DS < 0.15 are considered as ordered, residues with 0.15 ≤ DS < 0.25 are taken as flexible, whereas residues with 0.25 ≤ DS < 0.5 are considered as moderately disordered.”

Line 326 Please revise “only in several”

Answer: Thank you for the comment, we clarified in the text that there were only six cases of obtaining the crystal structures of almost full ATP synthase complexes.

Lines 342-349. The conclusion is a bit rudimentary. There are existing structural studies related to illumination, ionic strength etc. with a range of different techniques (including EM, SAXS and SANS) with direct information about the stacking, order, and in general the mesoscale structure of the thylakoids. They may or may not agree with your proposed hypothesis. Please perform a detailed critical discussion about how your hypothesis may fit into the framework of prior published results.

Answer: We thank the reviewer for this valuable comment. We significantly expand the conclusion and modified the text to make the conclusion clearer:

“We acknowledge that the presence of I-shaped dimers of chloroplast ATP-synthases is a hypothesis, which, ideally, should be directly checked, but one should be very careful about samples preparation and accurately control the ionic strength of the solution (e.g., for electron microscopy studies). Our structural data obtained by SAXS, though SAXS is an indirect method, point towards the possible presence of such dimers in purified samples of the cFOF1, which integrity was proven by BN-PAGE.

In literature, there are debates about the possible physiological roles of dimers and higher oligomers of cFOF1 [3,51]. Some papers show evidence of the presence of dimers of cFOF1 in microorganisms [19,20,52]. Other papers claimed that these dimers might be only aggregates without specific structural arrangement or functional role [51,53]. Commonly, inhibition of ATP hydrolysis in chloroplasts occurs via a redox switch inserted into a γ-subunit of cFOF1 [12], however, it is not excluded that there might be another way of ATP synthesis/hydrolysis regulation.

A formation of I-shaped dimers might establish an indirect interaction between lamellae (especially close to grana) and help to stabilize thylakoid stacks. Such dimers might be observed in more or less native conditions by cryo-electron tomography (Figure 4) and even the distance between neighboring lamellae is, in principle, allows such interaction. However, reported structural studies of chloroplasts were made during the light phase [3,54], leaving a possibility to find native I‐shaped dimers of cFOF1 in similar experiments at dark. When studying the mesoscale of thylakoids [3], I-shaped interactions could be indistinguishable from monomeric ATP synthases next to each other on adjacent lamellae. Of course, our hypothesis requires direct experimental verification, where the focus of the research will be directed specifically to the study of the protein supramolecular organization of lamellae and the search for mechanisms for their possible stabilization, as well as cryo-EM experiments with a focus on dimers or pairs of chloroplast ATP synthases.

Our study is intended to draw attention to a possible new type of ATP synthase dimerization, which has not been described previously, because regardless of the nature of such an interaction, we obtained experimental evidence of its presence (SAXS data), and given the fact that the ionic strength is comparable in order of magnitude in vitro and in vivo, especially given the fluctuations in ionic strength in plant chloroplasts during the light/dark cycles [5,6], we hypothesize that I-shaped dimerization may be reflected in native conditions and play a biological role, such as stabilizing thylakoid stacks, or regulating ATP synthesis/hydrolysis in plants.”

Figure S2 legend. Please use subscript in cFOF1

Answer: Thank you for this comment, we corrected the misprint.

Figure S2 and S3 are cited in reverse order, please rearrange.

Answer: We changed the order of the Figs. S2 and S3.

Comments on the Quality of English Language

The quality of English Language is relatively good. Some sentences should be clarified as detailed in the list above.

Answer: We checked the text and tried our best to improve it in terms of grammar and scientific terms.

Round 2

Reviewer 1 Report

No comments

Reviewer 2 Report

Dear Ms. Drugan

The authors clearly answered my questions and carefully addressed my concerns. I think the article is very well fitting now for being published in the International Journal of Molecular Sciences.

I would like to ask the authors to implement these minor changes:

The resolution of Figure 1 is not sufficient, please improve it.

In line 286 „behavior of the I(Q) SAXS profile (see the Fig. 1d).” I think the authors wanted to write Figure 1b. Please correct.

Lines 281-289 exclude the fact that the presented data is too noisy to state if there is a local maximum present. I suggest mentioning this to in this paragraph.

In line 388: „neighboring lamellae is, in principle, allows” needs to be corrected for grammar.

Thank you for the interesting article and for the expanded SI. It will be a useful reference for the community.

Sincerely,

Good